# Transcriptome Analysis of *Durusdinium* Associated with the Transition from Free-Living to Symbiotic

**DOI:** 10.3390/microorganisms9081560

**Published:** 2021-07-22

**Authors:** Ikuko Yuyama, Naoto Ugawa, Tetsuo Hashimoto

**Affiliations:** 1Graduate School of Science and Technology for Innovation, Yamaguchi University, 1677-1 Yoshida, Yamaguchi 753-8512, Japan; 2Faculty of Life and Environmental Sciences, University of Tsukuba, 111 Tennodai, Tsukuba, Ibaraki 305-8577, Japan; ugawa.naoto.sw@alumni.tsukuba.ac.jp (N.U.); hashimoto.tetsuo.gm@u.tsukuba.ac.jp (T.H.)

**Keywords:** RNA-seq, scleractinian coral, Symbiodiniaceae, endosymbiosis

## Abstract

To detect the change during coral–dinoflagellate endosymbiosis establishment, we compared transcriptome data derived from free-living and symbiotic *Durusdinium*, a coral symbiont genus. We detected differentially expressed genes (DEGs) using two statistical methods (edgeR using raw read data and the Student’s *t*-test using bootstrap resampling read data) and detected 1214 DEGs between the symbiotic and free-living states, which we subjected to gene ontology (GO) analysis. Based on the representative GO terms and 50 DEGs with low false discovery rates, changes in *Durusdinium* during endosymbiosis were predicted. The expression of genes related to heat-shock proteins and microtubule-related proteins tended to decrease, and those of photosynthesis genes tended to increase. In addition, a phylogenetic analysis of dapdiamide A (antibiotics) synthase, which was upregulated among the 50 DEGs, confirmed that two genera in the Symbiodiniaceae family, *Durusdinium* and *Symbiodinium*, retain dapdiamide A synthase. This antibiotic synthase-related gene may contribute to the high stress tolerance documented in *Durusdinium* species, and its increased expression during endosymbiosis suggests increased antibacterial activity within the symbiotic complex.

## 1. Introduction

The dinoflagellate of the Symbiodiniaceae family live symbiotically with a variety of marine invertebrates, including clams, sea slugs, sea anemones, foraminifera, and corals [1,2,3]. Among these, the symbiotic relationships between symbiodiniacean algae and cnidarians have been studied extensively. Symbiotic algae provide photosynthetic products to corals and receive nitrogen in exchange [4,5,6]. Published evidence indicates that the activity of symbiotic Symbiodiniaceae is under the control of the host corals [7,8,9]. In coral cells, algae are present in host-derived acidified vesicles that have carbon-concentrating mechanisms and activate photosynthetic capacity [7]. A recent transcriptome analysis found that dinoflagellate genes involved in molecular chaperoning as well as sugar and ammonia transportation were suppressed during the establishment of endosymbiosis with *Aiptasia* and coral planula larvae [10,11]. Gene expression analyses of actin, Ca^2+^ ATPase, and H^+^ APTase in Symbiodiniaceae also revealed that their expression patterns differed considerably between the non-symbiotic and symbiotic states [12,13]. However, despite these recent advances, many aspects of the changes dinoflagellate undergo during coral endosymbiosis establishment remain unclear. Previous studies have established a model endosymbiosis system consisting of monoclonal alga and juvenile corals, and transcriptome data for these coral–alga complexes have been published, with a focus on coral gene expression [11,14,15,16]. By contrast, the gene expression levels of dinoflagellate during coral endosymbiosis have not been investigated due to a lack of transcriptome data for the non-symbiotic state. Since the coral–alga model system facilitates the investigation of dinoflagellate gene expression and proliferation processes over time, it is well suited for examining changes in dinoflagellate during endosymbiosis.

In this study, we obtained transcriptome data for cultured *Durusdinium* that were previously used in an infection experiment with juvenile corals [14]. Therefore, in order to comprehensively investigate the changes in Symbiodiniaceae associated with the transition to the symbiotic state, we attempted to detect differentially expressed genes between the free-living and endosymbiotic states. Following the inoculation of juvenile corals with *Durusdinium*, its rate of increase was greater than that of *Cladocopium*, with about 300 *Durusdinium* cells per polyp detected by the 10th day of endosymbiosis, and about 600 cells per polyp detected by the 20th [14]. Here, we identified and functionally annotated differentially expressed genes (DEGs) between the non-symbiotic and symbiotic states, and performed phylogenetic analyses for a part of the DEGs to confirm that the DEGs were derived from *Durusdinium*.

## 2. Materials and Methods

Symbiodiniaceae strains CCMP 2556 (genus *Durusdinium*) were purchased from the Bigelow Laboratory for Ocean Sciences (West Boothbay Harbor, ME, USA; https://ccmp.bigelow.org/ (accessed on 20 Octorber 2011)). Cultures were grown at 24 °C under a 12 h/12 h light/dark cycle at 80 μmol m^−2^ s^−1^, in 100 mL of filtered seawater. Further details on RNA isolation and sequencing are provided in Appendix A.

Illumina HiSeq2000 transcriptome data for the symbiotic state of *Durusdinium trenchii* were obtained from the DDBJ Sequence Read Archive (accession nos. DRR119964-119967). Data obtained 10 and 20 days after coral incubation with *D. trenchii* [14], and those obtained in the present study, were used for DEG analysis. The quality checking of filtered reads, mapping, and the detection of DEGs between the non-symbiotic and symbiotic states were performed as described by Yuyama et al., 2018, and the supplementary materials. To confirm the calculated RNA-seq results, we performed bootstrap resampling of the raw read data, with 100 replicates per sample using the isoDE2 package [17], and with 100 replicates (*n* = 10) for each free-living or symbiotic sample (duplicate) in order to examine the expression changes between these states. Since the ANOVA using 100 bootstrap resampling results showed considerable variation among replicates for the data collected at day 10 of endosymbiosis, we eliminated these data from our analysis in order to detect changes related to endosymbiosis. Student’s *t*-test using 100 resampling results generated undetectably low *p*-values for most genes, so 10 items of data were randomly selected from the 100 replicates (100 bootstrap resampling replicates) and a *t*-test was performed. Differences in the mean (among the 100 replicates) expression of each gene between the two states were detected using the *t*-test (*p* < 0.025). The *p*-value was set to correspond to the number of DEGs detected in edgeR analysis (q = 0.01). These obtained DEGs were compared with those detected via the *TCC* analysis, and those in common were selected. We also selected genes with an expression change of log_2_ (fold change) > 1 between states. The functional annotations of the DEGs are described in the Appendix A.

Among the DEGs, we focused on dapdiamide A synthetase genes characteristic of *Dursdinium*. A phylogenetic analysis of dapdiamide A synthase was performed using the homologous gene sequences derived from diverse organisms in order to confirm that the genes were derived from *Durusdinium* rather than from bacteria. Further details on the phylogenetic analysis are given in the Appendix A.

## 3. Results and Discussion

In this study, we attempted to clarify the gene expression changes taking place in dinoflagellate during coral–alga endosymbiosis establishment. We prepared and sequenced a cDNA library derived from the symbiont culture, which isolated 25,068 contigs containing ORFs. Low reads derived from algae engaged in endosymbiosis with corals and free-living cultured algae were mapped against these contigs, and DEGs between these states were identified. The edgeR analysis identified 8543 DEGs, representing 34% of the candidate alga-derived transcripts. To validate these results, we used bootstrap replicates of RNA-seq data to detect DEGs between the non-symbiotic (*n* = 2) and symbiotic states (*n* = 2). Differences in the mean (among 100 × 2 replicates) expression of each gene between the two states were detected using the Student’s *t*-test (*p* < 0.025). The *p* value was set to correspond to the number (8642) of DEGs detected in the edgeR analysis (q = 0.01). A total of 4587 DEGs were common between both groups. Finally, we selected 1214 genes with log_2_ (fold change) > 1 between the free-living and symbiotic states in the edgeR analysis (Appendix A). The top 50 genes showing expression changes due to endosymbiosis with the lowest FDR included ribosomal proteins, heat-shock proteins, and chlorophyll-binding proteins (Appendix A). We also searched for GO molecular function terms that were enriched in these 1214 genes (Appendix A). The most enriched GO terms for these upregulated genes included protein–chromophore linkage and photosynthesis. The upregulated DEGs indicated that algae have enhanced photosynthetic activity during endosymbiosis with corals, which is consistent with previous reports of endosymbiosis in Symbiodiniaceae [7]. Seven processes were related to downregulated DEGs, including microtubule-based processes, mRNA splicing via spliceosome, and protein folding. Genes with decreased expression in microtubule-based processes include genes encoding tubulin, which is a component of flagella. This result may reflect the fact that algae lose their flagella inside corals [18]. Furthermore, a large number of ribosomal and chaperone proteins were detected among the downregulated DEGs, suggesting that some translational and protein-folding functions were inactivated following endosymbiosis establishment. Decreased expression of the chaperone gene has also been reported in the genera *Symbiodinium* and *Cladocopium* [11], and may represent a typical response to coral endosymbiosis establishment. It should be noted, however, that some of the DEGs detected include genes that were altered due to environmental differences between the two states. The time of year when the symbiotic and non-symbiotic states were cultured, as well as changes in the light environment, salinity, and CO_2_ in the coral cells, may have affected the genes whose expressions were altered. In order to investigate more specific changes in a symbiotic organism, we must more closely replicate the exact conditions of the culture strain and the symbiotic state.

Among the top 100 DEGs, two genes encoding dapdiamide A synthase (TRINITY_DN38519_c0_g1_i5.p1 (Figure 1) and TRINITY_DN38519_c0_g1_i1.p1) were found to be upregulated. Dapdiamide A synthase adds valine to the carboxylate of fumaramoyl-DAP to form dapdiamide A, an antibiotic, in *Pantoea agglomerans* [19]. Few studies have reported on the antibiotic synthase in Symbiodiniaceae; however, a recent large-scale transcriptome analysis identified dapdiamide A synthase in *Symbiodinium* [20]. Therefore, we performed a phylogenetic analysis to investigate whether the gene encoding dapdiamide A synthase is derived from Symbiodiniacea or from bacteria (Figure 2). One of the genes (TRINITY_DN38519_c0_g1_i5.p1) encoding dapdiamide A synthase was used for a BLASTp query against the NCBI database, and 117 sequences were selected for phylogenetic inference. The distribution of eukaryotic dapdiamide A synthase was restricted to large phylogenetic groups including stramenopiles, haptophytes, and alveolates (Symbiodiniaceae). In the ML tree (Figure 2), most of the eukaryotic sequences formed two separate clades, A and B. Clade A comprises sequences derived from bacillariophytes (stramenopiles), haptophytes, and Symbiodiniaceae. Nine Symbiodiniaceae sequences were monophyletic, with 86% support, and its sister group was shared by *Emiliania huxleyi* (haptophyte) sequences. These branching patterns suggest that the eukaryote–eukaryote lateral gene transfer of dapdiamide A synthase occurred between *Emiliania* and Symbiodiniaceae. In addition, close relationships between two *Symbiodinium* sequences as well as archaean (100%) and *Aureococcus* (pelagophyte) (no support) sequences on the lower part of the tree suggest other types of lateral gene transfer involving Symbiodiniaceae. However, these genes were detected only in the genera *Symbiodinium* and *Durusdinium* of Symbiodiniaceae in this study. *Durusdinium* species exhibit high stress resistance, and have been reported to confer this property to their host corals [21]. Our results show that both dapdiamide A synthase genes from *Durusdinium* were upregulated during endosymbiosis establishment, which may enhance antibacterial action and confer stress tolerance to the host coral.

In this study, *Durusdinium* genes that exhibited expression changes during coral endosymbiosis establishment were selected using two analysis methods for functional analysis. The weakness of this study is the small number of replicas and the different timings of the fixation of symbiotic and non-symbiotic dinoflagellate. In future gene expression research, it will be necessary to improve these areas. In addition, transcriptome data do not necessarily correlate with protein expression data, thus requiring proteome analysis to elucidate the entire internal symbiotic process. The roles of these genes in dinoflagellate adaptation to the host coral environment need to be further investigated; such data could be useful in clarifying the evolutionary process of symbiont trait acquisition.

## Figures and Tables

**Figure 1 microorganisms-09-01560-f001:**
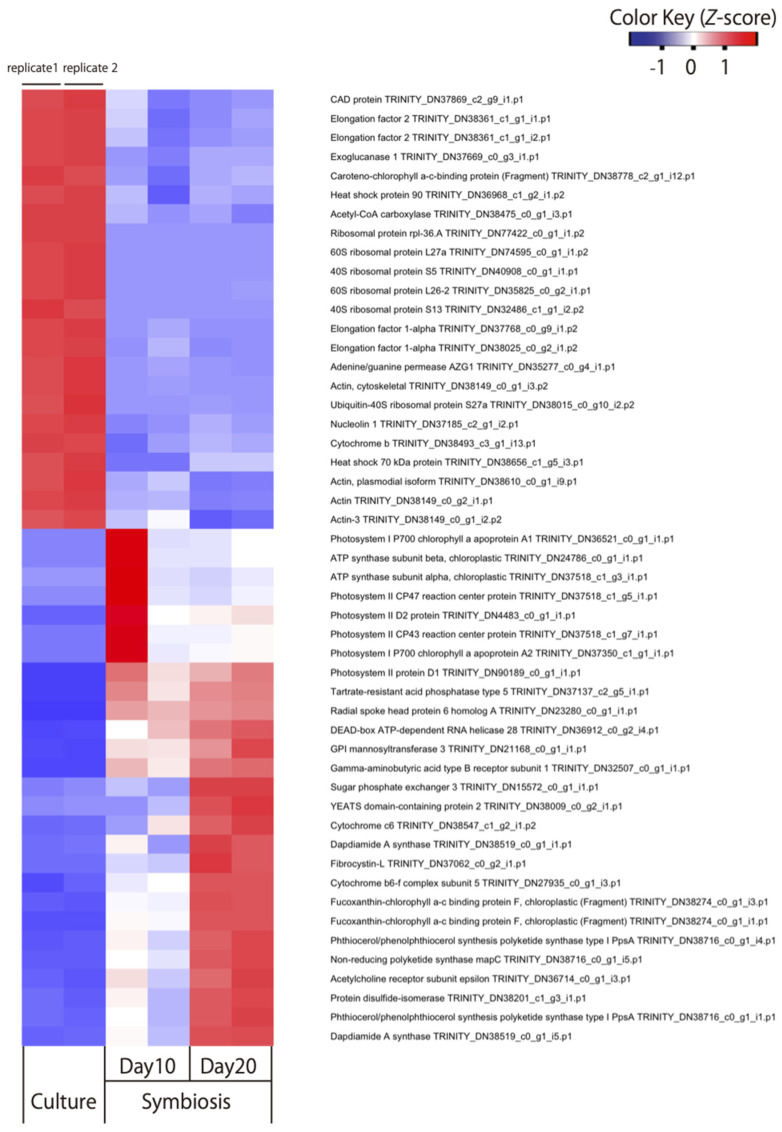
Heat map of RNA-seq analysis for 50 selected genes showing different gene expression pattern in free-living and symbiotic *Durusdinium* at 10 and 20 days post-inoculation (*n* = 2). Among the 1214 DEGs shown in Appendix A, the 50 genes with the lowest false discovery rate are summarized in the heatmap.

**Figure 2 microorganisms-09-01560-f002:**
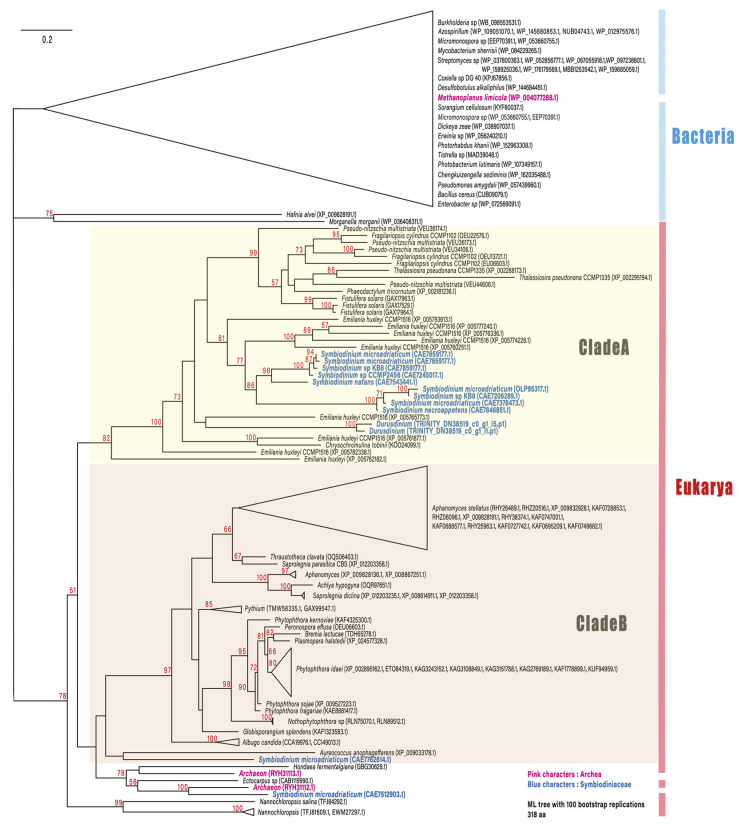
Maximum likelihood phylogenetic tree describing the relationships among dapdiamide A synthase proteins from representative eukaryotes and prokaryotes. All bacterial and one archaean sequence were separated from the remaining eukaryotic/archaeal sequences (78%). Most of the eukaryotic sequences formed two separate clades, A and B. In clade A, bacillariophyte sequences were monophyletic (99%), excluding haptophyte and Symbiodiniaceae sequences. Nine *Symbiodinium* sequences were monophyletic, with 86% support, and its sister group was shared by *Emiliania huxleyi* (haptophyte) sequences. Numbers in red near the nodes are ultrafast bootstrap support values; values < 50% are not included. Blue indicates genes derived from Symbiodiniaceae.

## Data Availability

The raw fastq files for the RNA-seq libraries were deposited at *DDBJ* Sequence Read Archive (DRA) with the accession number of DRA10343.

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
