# Peer review of "Transcriptome Analysis of Durusdinium Associated with the Transition from Free-Living to Symbiotic"

_microorganisms, 2021, doi:10.3390/microorganisms9081560_

Round 1

Reviewer 1 Report

Yuyama et al. reported gene expression levels of symbiotic algae in coral before and after establishment of symbiosis. They used a unique method by comparing non-symbiotic (free-living) states and symbiotic states (but just after 10-20 days from inoculation). They clearly showed transition of gene expression during establishing symbiosis and these results can be important data set to clarify symbiosis establishment for not only coral-algal relationship but also other animal-algae symbiosis. It’s interesting that the number of down-regulated genes was much higher than the number of up-regulated genes during symbiosis establishment.

They validated their results by using boot strap resampling technique and made up for low numbers of replicates. Trend of gene expression response in their results looks common between most of replicates such as Fig.2. Not only reporting data set, but also publishing this methodology is valuable.

There are still a few reports of gene expression patterns related to symbiosis from algal side and the number of reports such as this study should be increased.

For these reasons, I think this paper can be published after some minor correction. Please see my comments below.

L19         “Durusdinium” and “Symbiodinium” should be in Italic.
L50-51    “2+” and “+” should be superscript.
L68         “600 Drusdinium” means “600 cells of Drusdinium”?
Also, “600 Durusudinium detected in corals” indicates that 600 cells were detected in total from multiple corals? Or, is it mean number from one individual? Should show carefully.
And, it’s better to show cell number of 10 days also as author present gene expression results of 10 days in Fig.2.
Typo: Drusudinium should be Drusdinium

L125-132 I couldn’t understand how authors conducted this analysis, especially “100 replicates” in L128 means. If they do random selection of 10 data from 100 resampling data, I thought replicates were 10. Did they 10 times selection of 10 data from 100 resampling data?
“T-test”, “t-test”, “Student’s t-test” are same meaning? If so, “Student’s t-test” should be shown in first and then use “t-test” from second, but should not use “T-test”.

L193       “198” should be “189”.
L209  What is “(F)”? Is it need?
L222      “198” should be “189”. Total number cannot be 1,214.
L232 What is “Val”? Is it common word?
L277 Typo: Drusudinium should be Drusdinium

About figures:
Resolution of figures is low. I think reader cannot identify gene or species names (e.g. Fig.2, Fig.4). Should prepare higher resolution figure.

Reviewer 2 Report

In this work, the authors compared the gene expression profiles of two new culture (i.e., free-living)-based libraries vs. a published endosymbiotic (in hospite) dinoflagellate RNA-Seq dataset from a prior work. They then used two different statistical approaches to quantify differential gene expression. This is a very small dataset, and 2/3 of the data are from another work. The novel data represents a sample size of two, and many of the differences could be attributed to things like sampling year, season, culture environment between the coral and the flask, etc. Therefore, I think the only way this work could be publishable is as a very short and small note in which the authors admit that many of the changes in gene expression could be due to not having undertaken the analyses at the same time under the same conditions. Ideally, you would culture corals or anemones alongside the dinoflagellate cell cultures, harvest at the same time, temperature, light level, CO2 level, etc., and THEN you can ascribe differences to the symbiotic state (free-living vs. in hospite). Otherwise, you can only capture a portion of the variation by doing such a meta-analysis against prior works. If this work could be shortened to about 1,000-2,000 words with 1-2 figures, a very brief Introduction, and a Discussion that acknowledges these deficiencies, I think this could be publishable in Microorganisms. The better alternative, though, would be to do a properly designed, side-by-side study, in which you could actually attribute differential gene expression to lifestyle, as opposed to other experimental factors. Therefore, I would deem this a major revision if the authors choose to convert this into a note. Otherwise, I encourage them to carry out a larger study, with the goal of generating a larger (in scope) work on the molecular basis of endosymbiosis in the future.

Major comments

  1. This work must be proofread by a native English speaker prior to its resubmission. Even the title doesn’t make sense (though I can guess the meaning: a shift from a free-living to an endosymbiotic lifestyle). I have pointed out a few of the errors below, but please note that it does not reflect an exhaustive list.
  2. See my comment in the Summary. If you want to attribute changes in gene expression to differences in lifestyle (culture vs. in hospite), you’ll need to account for all sources of variation in gene expression, especially light, temperature, and things of that nature. This is the advantage of using anemones, in fact, since they can be cultured under similar conditions as the dinoflagellates. As for now, many of the DEGs likely stem from differences in light, temperature, salinity, food availability, etc., and not necessarily being within or outside of coral gastrodermal cells. If the sample size was larger, maybe 5-6 samples of free-living vs. in hospite, you may be able to make larger claims about free-living vs. endosymbiotic, but a 2 vs. 2 design carried out years apart will not lend much strength to the conclusions. This is why I suggest above an alternative in making this a very brief note to emphasize the preliminary nature of the study. Then, you could use it to launch a bigger, more properly replicated and conducted study in the coming years (or for a grant submission).
  3. If you actually want to do enrichment, you need to determine whether the DEGs are over-represented versus their abundance in the transcriptome. For instance, if many cytoskeleton genes are found in the genome and many cytoskeleton DEGs are identified, then they may not actually be “enriched.” It might be more of a semantics issue. For instance, if 20% of the sequenced genes are involved in the cytoskeleton, and 20% of the DEG pool genes are cytoskeleton genes, that is an interesting result, but it is not “enriched” with cytoskeleton genes since they are found in a similar proportion as in the global transcriptome. Instead, you are simply reporting the “most common pathways” uncovered. I advise doing the enrichment analysis, but that will take a much more thorough annotation of the reference transcriptome. Otherwise, I would change the wording to “most represented pathways” or something along those lines.
  4. Lines 287-290: There is actually a rich literature from Taiwan on the important of RNA splicing in Symbiodiniaceae molecular biology. I would recommend consulting these works (which also looked at the proteins).
  5. Speaking of which, be careful in over-interpreting the gene expression-based findings since there is no correlation between gene expression and concentration of the respective protein in Symbiodiniaceae. The respective proteins MIGHT play a role in symbiosis, but you would need to verify that with western blots or proteomics. Otherwise, it is just as likely that the differentially concentrated protein pool is comprised of very different molecules. For these reasons, I would recommend a proteomic approach in the future.

Minor comments

Minor comments-overall

  1. It should be “endosymbiosis” or “endosymbiotic” (not “symbiosis” and “symbiotic,” respectively). Similarly, it should be “dinoflagellate” instead of the less specific and more antiquated term “algal.”
  2. Genus and species names should be in italics. As of right now, it’s a mix of regular font and italics.
  3. “Symbiodiniaceae” is often misspelled (see line 290, for instance).
  4. Line 367: actin?
  5. There is a mix of many different fonts. I would choose one style and maintain consistency throughout.

Minor comments-Introduction

  1. I think rather than state what you found at the end of the Introduction, I would pose hypotheses that you aimed to test. I am guessing you expected to see differential gene expression between the free-living and endosymbiotic states, right?

Minor comments-elsewhere

  1. RNA isolation and transcriptome sequencing (no comma)
  2. It is not possible to read the gene names in the heat map. Maybe you can reduce it to the top 50 DEGs?
  3. My favorite part of this paper is having the two-tiered method for selecting DEGs versus the more traditional approach of having a single method. I would advise sticking with this approach once you are able to beef up the sample size.
